# Latent Class Analysis to Identify Parental Involvement Styles in Chinese Children’s Learning at Home

**DOI:** 10.3390/bs12070237

**Published:** 2022-07-15

**Authors:** Xiaorui Huang, Randall E. Schumacker, Bin-Bin Chen, Ming-Ming Chiu

**Affiliations:** 1Collaborative Innovation Center of Assessment toward Basic Education Quality, Beijing Normal University, Zhuhai 519087, China; xrhuang@bnu.edu.cn; 2College of Education, The University of Alabama, Tuscaloosa, AL 35487-0231, USA; 3Development of Psychology, Fudan University, Shanghai 200433, China; 4Assessment Research Center and Department of Special Education and Counseling, The Education University of Hong Kong, Hong Kong, China; mingchiu@eduhk.hk

**Keywords:** latent class analysis, parental involvement styles, academic achievement, Chinese students

## Abstract

Background: Parental involvement is one of the most important factors affecting students’ academic learning. Different families seem to show similar parental involvement patterns. This study employed a representative sample of 12,575 seventh- and eighth-grade Chinese students’ parents to explore the patterns of parental involvement. (2) Methods: Latent class analysis (LCA) was used to identify different parental involvement styles in children’s studies at home. Discriminant analysis, MANOVA, post-hoc tests, and effect size were used to verify the LCA results. (3) Results: Four distinctive latent class groups were identified and named: supportive (20%), permissive (54%), restrictive (8%), and neglectful (18%). A discriminant analysis supported the LCA group classification results. The MANOVA results indicated statistically significant differences between the four latent classes using the set of predictor variables. The post-hoc test results and effect sizes showed that the predictor variables had substantial differences among the four latent class groups. Parental education and family income showed statistically significant links to these four parental involvement styles, which, in turn, were linked to students’ academic achievement according to the MANOVA, effect sizes, and post-hoc test results. (4) Conclusions: Parental involvement styles in children’s learning at home can be identified and categorized into four different latent class styles.

## 1. Introduction

“The differences among families seem to cluster together in meaningful patterns”—Lareau (2011, p. 3)

Parent personality, educational beliefs, parenting styles, and parent–child interactions affect parental involvement, which can affect children’s academic learning [1]. However, most past studies ignored differences across profiles of parents within a population (assuming homogeneity) and only considered linear or quadratic relations between parental involvement behaviors and their children’s academic learning. These assumptions can distort the relations between parental involvement and children’s academic learning.

Hence, this study aims to extend the results of past studies on parental involvement in children’s learning at home in two directions: (a) specifying profiles of parental involvement in children’s learning at home and (b) determining these parental involvement profiles’ antecedents (parent education and family income) and consequences (students’ academic achievement). This study used latent class analysis (LCA) on data from 12,575 seventh- and eighth-grade Chinese students’ parents to explore their patterns of parental involvement in their children’s learning at home.

### 1.1. Definitions of Parental Involvement

Parental involvement is defined as a parent’s engagement in activities related to his or her child’s academic achievement (parental involvement has been used interchangeably with parent engagement, parent participation, home literacy practices, home experiences, parent–school relations, parent tutoring, parent curriculum, and parent–child reading [2]). Regarding their children’s education, many parents act as guardians, teachers’ assistants, and voluntary tutors [3]. Parental involvement includes home-based involvement and school-based involvement [1,4,5]. Home-based parental involvement includes parent education and parental involvement in children’s learning at home [6]. Specifically, regarding support for their child’s learning, parental involvement at home includes homework-related supervision (e.g., supervision of schoolwork, checking homework, and homework assistance), reading with children, education expectations and aspirations, and parental attitudes toward education [7,8,9]. School-based involvement includes communication between parents and schools, parents’ voluntary participation in schoolwork, and parent participation in school policy decision making [4]. The present study focused on home-based involvement.

### 1.2. Inconsistent Relation between Parental Involvement and Academic Achievement

Previous studies showed mixed effects of parental involvement on student learning outcomes. Some studies showed that parental involvement was positively related to students’ academic achievement [10,11,12,13,14,15]. The effect sizes between parental involvement and academic achievement were found to be 0.25 to 0.30 [11,16], 0.22 to 0.62 for minority groups [12], 0.52 for Latino students [17], 0.29 for kindergarteners and primary school students, and 0.12 to 0.35 for secondary school students [10,16]. In addition, Wilder performed a meta-synthesis across meta-analysis studies and concluded that parental involvement had a positive impact on children’ academic achievement overall [18].

However, other studies showed nonsignificant or even negative links with students’ academic achievement. For example, home-based involvement had very little effect (*r* = 0.03) on students’ academic achievement, no significant effect according to the data of the National Educational Longitudinal Study of 1988 (CI was −0.17 to 0.08), and a negative effect (CI was −0.14 to −0.11) when using the Longitudinal Study of American Youth data [19]. Mattingly et al. evaluated different programs of parental involvement from kindergarten to secondary education and found no evidence of a connection between promoting parental involvement and improvement in students’ learning [20]. Hill and Tyson found a positive relation between parental involvement and academic achievement, but the effect size was small (0.04) [19]. Taking these results together, the relation between parental involvement and students’ achievement is inconsistent.

Different definitions of parental involvement (typically, specific behaviors or activities) across studies might account for these inconsistent results. Helping with homework had a negative effect on academic achievement (*r* = −0.11) [19], while supervising homework had little effect on students’ academic achievement (*r* = 0.02) [10]. Listening to children read had a much greater positive effect on students’ academic achievement (d = 0.51) than reading to them (*r* = 0.17; d = 0.18) [10,21]. Among different parental involvement behaviors, parent expectations also showed a positive link with students’ academic achievement (*r* = 0.33; *r* = 0.22) [10,11]. In addition, parent–child interactions could improve a child’s learning outcomes [22]. Overall, the relation between parental involvement and students’ academic achievement varied according to the definition of parental involvement in specific studies.

### 1.3. Profiles of Parental Involvement in Students’ Learning at Home

To determine how parental involvement is related to children’s learning outcomes, we can evaluate the effects of overall parental involvement or of each of its components on children’s academic achievement, but this variable-centered approach cannot address the complex interactions among the variables. As the number of variables increases, the number of higher-order interactions (three-way, four-way, etc.) increases, which reduces the statistical power to detect significant effects.

Furthermore, the variable-centered approach assumes a sample from a single population, so its analytic results (e.g., reading to children at home is positively correlated with their reading performance) are assumed to be generalizable to that entire population [23]. When the sample has multiple distinct subpopulations, the relations can differ in size or even have opposite directions across subpopulations. Hence, differences in subpopulations can contribute substantial error to variable-centered studies. Hence, family involvement studies need a method for identifying subpopulations [24].

To overcome this limitation of a variable-centered analysis, we can instead use a person-centered analysis. Specifically, LCA can be used to identify how dimensions are organized within individuals and how values along these dimensions are distributed across a heterogeneous population to separate it into more homogeneous subpopulations [23,25]. Then, we can describe the characteristics of each subpopulation and study their antecedents and consequences [25].

### 1.4. The Present Study

This study addresses three issues. First, we integrated the indicators of parental involvement used in empirical studies. The LCA identified groups (styles) of parental involvement in learning at home to identify different subpopulations (i.e., classes) of parents based on their levels of parental involvement behaviors. Second, past studies showed that parents’ education, income, or social class can affect parental involvement [1,6,26]. Therefore, we examined whether these factors were linked to profiles of parental involvement. Third, as past studies showed inconsistent links between parental involvement and academic achievement, we examined whether these profiles of parental involvement were related to academic achievement.

## 2. Methods

### 2.1. Participants

The data used in the analysis were reported by parents. Stratified sampling yielded a representative sample of 12,575 7th- and 8th-grade students in a city in central China. This city’s seventeen administrative areas were clustered into four groups with a similar gross domestic product (GDP), education spending, mean score on high-school entrance examinations, urban–rural distribution, resident population, and total population. From each of the four groups, we selected 53 junior high schools based on school type, school size, school educational quality, and the number of schools within that area. Within each of these schools, we sampled two or three 7th- and 8th-grade classes with diverse academic achievement levels (according to midterm test scores and class labels, such as low- and high-ability classes and/or talented classes). All of the students and their parents in the sampled classes participated in this project, resulting in a total sample of 12,575 students (53% boys and 47% girls; 52% 7th-grade students and 48% 8th-grade students) and parents. The sampled students’ parents voluntarily responded to questions on a questionnaire.

### 2.2. Measures

We used 17 indicators of parents’ involvement behaviors during their children’s learning at home from past studies and organized them into 5 dimensions (see the descriptions in Table 1 and correlations in Table 2).

#### 2.2.1. Involvement in Children’s Homework at Home

These items were adapted from Cheung and Pomerantz’s parental involvement scale [27]. It consists of 4 four-point Likert-type items (strongly disagree, disagree, agree, and strongly agree) about involvement in homework (1. I always examine my child’s homework; 2. I always supervise my child to ensure that they finish homework on time; 3. My child and I finish the assigned homework together, which requires a parent’s collaboration; 4. I assign extra homework for my child).

#### 2.2.2. Parent–Child Interactions

Four four-point Likert-type items (never do this, seldom do this, sometime do this, and always do so) about parent–child interactions at home (1. I always watch and discuss the TV programs; 2. My family members (including me) always buy extra-curricular books for my child; 3. I always listen to my child’s reading or I always read to my child; 4. I communicate the learning issues that my child meets) were included. These items implied that parents accompanied their children at home and had interactions with their children.

#### 2.2.3. Emotional Support

Emotional support has two aspects: positive emotional support and negative emotional support. Here, it consisted of 2 four-point-scale items (I tutor my child’s homework patiently; I get angry when my child doesn’t finish homework) and 4 binary items (I talk with my child heart-to-heart/I analyze the reasons for my child’s scores/I scold my child/I treat my child deliberately coolly when s/he scores poorly). The items regarding treatment of the child after achievement of poor scores were adapted from the item “I believe that scolding and criticism help my child” [28,29].

#### 2.2.4. Conflicts with Children

Two items were used to measure the conflicts with children. One binary item was used to measure whether parents had conflicts with their children and one four-point scale was used to measure how parents dealt with parent–child conflicts. Four choices were provided: 1. Criticize the child and force him/her listen to me; 2. Persuade my child to follow my opinion; 3. Discuss with my child and reach a consensus; 4. Completely follow my child’s opinion.

#### 2.2.5. Setting Rules for Children

One item on a four-point scale was used to measure whether parents set rules for their children: “I set the TV and game time for my child” (from “never” to “always do this”). This item was adapted from the item: “I have strict, well-established rules for my child” [29]. This item was adapted because it related to specific parents’ controlling behaviors.

#### 2.2.6. Parent Education, Family Income, and Academic Achievement

We also included parent education, family income, and academic achievement. Parent education was measured with the parents’ highest education levels. Academic achievement was measured with the math, Chinese, and English test scores. Students in the same grade and the same school used the same test, so we computed the standardized scores of each subject within each grade. The mean score from the three subject areas was computed and used as the academic achievement measure for each student. The percentage of missing data across variables ranged from 0.01% to 2.27%.

#### 2.2.7. Data Analysis Strategy

To identify subpopulations of parents with similar parental involvement, we compared latent class models of the indicator variables in Table 1 via Mplus 7.31 [30]. We estimated the missing data with full-information maximum likelihood [31]. The maximum likelihood estimation with robust standard errors was used to estimate the indicator variable parameters.

To determine the relative fit of different latent class models, we used the Bayesian information criterion (BIC) to obtain consistent estimates regardless of the sample size, unlike other information criteria [32]. Lower BIC values indicate a better fit [33]. We also used the Vuone–Lo–Mendell–Rubin likelihood ratio test (VLMR-LRT), the adjusted Lo–Mendell–Rubin likelihood ratio test (Adjusted LRT), and the bootstrapping likelihood ratio test (BLRT) to compare the models with C classes and C-1 classes [33]. A significance test (*p* < 0.05) of these three LRTs indicated whether a model with C classes could not be rejected in favor of one with C-1 classes. As shown by Nylund and his colleagues, the BLRT is more accurate than VLMR-LRT. Entropy values ranging from 0 to 1 indicated the conditional probabilities of individuals’ group membership [34]. A high value of entropy (≥0.80) indicated a confident classification of individuals and adequate separation between the latent classes [35], while a somewhat lower value (≥0.70) was marginal [36]. In short, we compared the following measures of each model: AIC, BIC, Adjusted BIC, VLMR-LRT, Adjusted LRT, BLRT, and entropy.

A discriminant analysis and multivariate analysis of variance (MANOVA) were used to further assess the accuracy of the LCA. Using the LCA’s statistically significant indicator variables, the discriminant analysis predicted the latent class probability of group membership [37]. Furthermore, the MANOVA used the latent class membership to predict the LCA’s statistically significant indicator variables. Post-hoc tests were used to determine the significance of indicator differences across latent class groups, and the effect sizes showed the practical differences in these indicators.

Analysis of variance (ANOVA) was then used to examine whether (a) explanatory variables (parent education or family income) were linked to the latent parental involvement classes and (b) whether the latent parental involvement classes were linked to students’ academic achievement.

## 3. Results

Table 1 reports the means and standard deviations of each parental involvement item (see the correlations in Table 2).

### 3.1. Latent Class Analysis

The latent class analysis tested five different models of one to five classes. Although the five-class model had the lowest BIC value (386,371), the four-class model had the highest entropy (0.760) and latent class probabilities exceeding 0.83, and the LMRT (LMR = 1654.60, *p* = 0.016) showed that the five-class model was not superior to the four-class model (*p* > 0.01); together, they indicated that the four-class model was the best model (see Table 3). These four classes were: permissive (54%), supportive (20%), neglectful (18%), and restrictive (8%).

### 3.2. Classification Accuracy

The discriminant analysis and MANOVA results supported the four-class model. The discriminant analysis indicated that the classification of group membership was 93% correct for permissive (91.1%), supportive (99.7%), neglectful (96.5%), and restrictive (79.0%; see Table 4). The MANOVA results showed that the latent classes significantly predicted all 17 indicator variables (F = 577.40, df = 51, *p* = 0.0001, η^2^ = 0.59, and power = 1.0; see Table A1).

The MANOVA post-hoc tests showed that 91% (93/102) of the indicators for the pairs of classes (permissive vs. supportive, permissive vs. neglectful, etc.) differed significantly (see Table A1 and Table A2). The three nonsignificant differences were for permissive versus supportive (action during conflict with child, *p* = 0.196) and permissive versus restrictive (supervising homework, *p* = 0.054; assigning homework, *p* = 0.371). In addition, some pairs also showed low effect sizes: permissive vs. supportive (angry about unfinished homework, ES = −0.10; scolding child for poor scores, ES = −0.09; talking with child about poor scores, ES = −0.06; conflict with child? ES = −0.08), permissive vs. restrictive (setting TV and game time, ES = 0.17), and supportive vs. neglectful (coolly treating poor-scoring child, ES = −0.11). Hence, these items were not well differentiatd within these pairs of parental involvement types.

### 3.3. External Validation of Latent Classes

Parent education differed significantly across the four latent classes (MANOVA results: F [3, 12489] = 394.0, *p* = 0.0001) and across all pairs (post-hoc comparisons all showed *p* < 0.0001), showing a clear ordering from highest to lowest with substantial effect sizes: supportive > permissive > restrictive > neglectful. Likewise, family income differed significantly across the four latent classes (F [3, 12206] = 121.8, *p* = 0.0001) and across all pairs (all post-hoc comparisons were significant), showing a clear ordering from highest to lowest with substantial effect sizes: supportive > permissive > restrictive > neglectful. In addition, students’ academic achievement differed significantly across the four latent classes (F [3, 12387] = 43.5, *p* = 0.0001) and across all pairs (all post-hoc comparisons were significant), showing a clear ordering from highest to lowest with substantial effect sizes: supportive > permissive > neglectful > restrictive.

Together, these results show somewhat similar orders.Parent education:       Supportive > Permissive > Restrictive > Neglectful.Family income:       Supportive > Permissive > Restrictive > Neglectful.Academic achievement:       Supportive > Permissive > Neglectful > Restrictive.

These results support those of past studies showing how parent education and family income are linked to parental involvement, along with its link to student academic achievement. As a result, these results help validate the four parental involvement classes.

### 3.4. The Relations between Indicators of Parental Involvement and Academic Achievement in Each Latent Class

For the four parental involvement profiles, the correlations between indicator and academic achievement showed mixed results (see Table A3). Furthermore, the correlations between an indicator and academic achievement could differ across parental involvement profiles. For example, the correlation between examining homework and academic achievement was significantly positive overall (0.13, *p* < 0.01), as it was for restrictive parental involvement (0.09, *p* < 0.01), it was but significantly negative for neglectful parental involvement (−0.06, *p* < 0.01) and nonsignificant for both permissive and supportive parental involvement. As these correlations differ across different latent classes of parental involvement, ignoring these distinct subpopulations yields incorrect, oversimplified results for a heterogeneous population.

## 4. Discussion

Across the last several decades, studies of the impact of parental involvement on student achievement have shown mixed results, in part because of different definitions of parental involvement [10,11,15]. This study showcased a methodology of LCA, discriminant analysis, MANOVA, and correlation analysis in order to determine and validate subpopulations of distinct types of parental involvement.

Specifically, we used LCA on 17 commonly studied measures of parental involvement to identify four latent classes of parental involvement in children’s home learning: permissive, supportive, neglectful, and restrictive. The discriminant analysis showed satisfactory accuracy for the classifications of permissive, supportive, and neglectful parental involvement (all exceeding 91%), but with less accuracy for classifying restrictive parental involvement (only 79%), indicating the need to improve the indicators for distinguishing it from the other three parental involvement styles. The mean effect sizes of three indicators—“I treat my child deliberately coolly when s/he scores poorly” (0.57); “I talk with my child when he/she scores poorly” (0.55); “Do you have conflicts with your child” (0.42)—were far below 0.80, indicating that they had unacceptably low power for distinguishing restrictive parental involvement from the other parental types of involvement [36]. Future studies can examine these items and, perhaps, revise them to improve their accuracy. The other indicators showed sufficient effect sizes and, thus, power for differentiating the four latent classes.

Supportive parental involvement had the highest values regarding engagement with children’s homework, parent–child interactions, emotional support, and setting clear rules for children at home, along with the lowest rates of conflict with children. These parents were highly responsive to their children, highly respected them, and made high demands of them (e.g., setting clear rules for children), similarly to the attributes of authoritative parenting [38]. Supportive parents had the highest parent education level and highest family income, and their children had the highest academic achievement. These results suggest that these parents had the knowledge, skills, and resources (e.g., to buy educational materials or learning opportunities/experiences) to effectively help their children learn more. Other studies suggested that their abilities and resources contributed to their greater efficacy in helping their children [39].

Permissive parents were highly responsive to their children and highly respected, them but made few demands of them [40]. Compared to supportive parents, permissive parents had substantially lower involvement, lower education levels, and lower income, while their children performed worse academically than those of supportive parents. In turn, permissive parents had more involvement, more education, and more income than both neglectful and restrictive parents, and children of permissive parents showed higher academic achievement than those of restrictive parents (not significantly differently from those of neglectful parents).

Restrictive parents placed high demands on their children, but were not responsive to them, consistently with the authoritarian parenting style [41]. The emotional support for children’s homework was lowest for this type of parental involvement. Although parent education and family income were higher for restrictive parents than for neglectful parents, academic achievement was lower for children of restrictive parents than for children of neglectful parents. These results are consistent with those of studies showing that restrictive parenting reduces children’s learning motivation and learning outcomes [42].

Neglectful parents had the lowest parental involvement in this study. They had low demands and low responsiveness to their children’s learning at home, similarly to a neglectful (or uninvolved) parenting style [38]. Parents in this latent class had the lowest parent education and lowest family income, though their children’s academic achievement was similar to that of the children of permissive parents and higher than that of the children of restrictive parents. These results suggest that high demands without emotional support harm children’s learning outcomes more than the absence of demands or support. Future studies can examine this issue further.

The correlations between indicators of parental involvement and academic achievement varied, sometimes with opposite directions, across latent parental involvement profiles. This result shows that the relation between such an indicator and academic achievement is not universal, but dependent on the parental involvement context [11,19]. Hence, ignoring the parental involvement context can yield inaccurate results [43,44].

## 5. Conclusions

The current findings showed that parental involvement styles in children’s learning at home can be identified and categorized into four different latent class styles. These results suggest that only by identifying and modeling parental involvement styles at home can researchers address research questions concerning parenting and students’ academic achievement.

## 6. Future Research

Future studies could include questions related to parental involvement with children that complete homework on electronic devices [45,46,47,48]. The use of the four parenting styles would also permit statistical analyses with the variables of parent gender, subject area, and course type. Furthermore, this study showed how parent education and family income (components of family social economic status (SES)) are related to parental involvement, so future studies can examine this relation and its possible causal mechanisms [6]. As this study indicates links between parental involvement and student academic achievement, future studies can discern its causal mechanisms and mediating variables, such as students’ emotions or parental attachment [49]. Future research can explore how parents’ interactions in schools and with teachers and parental involvement styles with their children mutually influence each other [50]. Future studies can also examine these relations in other countries. Clarification of the definitions of the parental involvement types delineated in this study will facilitate future research on parental involvement (e.g., comparisons across demographic attributes).

## Figures and Tables

**Table 1 behavsci-12-00237-t001:** Descriptive statistics on 17 indicator variables.

Item	Item Contents	Item Type	M	*SD*
1	I always examine my child’s homework. (Examine HW)	4-point	2.42	0.807
2	I always supervise my child to finish homework on time. (Supervise HW)	4-point	2.88	0.815
3	My child and I always finish the assigned homework together, which requires a parent’s collaboration. (Finish HW together)	4-point	2.64	0.984
4	I assign extra homework for my child. (Assign HW)	4-point	1.64	0.729
5	I tutor my child’s homework patiently. (Patiently)	4-point	2.43	0.915
6	I get angry when my child doesn’t finish homework. (Angry)	4-point	2.06	0.863
7	I read to my child/I listen to my child’s reading. (Read)	4-point	1.98	0.803
8	I always watch and discuss the TV programs with my child. (Discuss TV)	4-point	2.25	0.823
9	I set the TV and game time for my child. (Set time)	4-point	2.49	0.975
10	I communicate the learning issues that my child meets. (Communicate)	4-point	2.43	0.823
11	My family members (including me) always buy extra-curricular books for my child. (Buy books)	4-point	2.35	0.761
12	When you have a conflict with your child, what do you do? ^1^ (Conflict with child)	Binary	2.48	0.684
13	I scold my child when s/he scores poorly. (Scold child)	Binary	7.50%	
14	I treat my child deliberately coolly when s/he scores poorly. (Treat cool)	Binary	4.40%	
15	I talk with my child when s/he scores poorly. (Talk with child)	Binary	70.50%	
16	I help my child analyze the reason for why he/she scores poorly. (Help analyzing)	Binary	70.60%	
17	Do you have conflicts with your child? (Conflict with child)	Binary	63.20%	

^1^ Note: Four choices were provided: 1 = criticize the child and force him/her to follow; 2 = persuade my child to follow my idea; 3 = discuss with my child and reach a consensus; 4 = completely follow my child’s opinion.

**Table 2 behavsci-12-00237-t002:** Correlation matrix (17 indicator variables) ^1^.

	1	2	3	4	5	6	7	8	9	10	11	12	13	14	15	16
2	0.424 **															
3	0.415 **	0.353 **														
4	0.309 **	0.196 **	0.247 **													
5	0.413 **	0.260 **	0.425 **	0.273 **												
6	0.041 **	0.167 **	0.023 **	0.073 **	−0.035 **											
7	0.323 **	0.197 **	0.312 **	0.267 **	0.346 **	−0.009										
8	0.177 **	0.158 **	0.229 **	0.152 **	0.274 **	−0.017	0.332 **									
9	0.191 **	0.235 **	0.241 **	0.181 **	0.226 **	0.092 **	0.189 **	0.262 **								
10	0.373 **	0.256 **	0.408 **	0.281 **	0.469 **	−0.014	0.376 **	0.348 **	0.301 **							
11	0.235 **	0.163 **	0.261 **	0.238 **	0.270 **	−0.026 **	0.245 **	0.235 **	0.213 **	0.332 **						
12	0.124 **	0.046 **	0.156 **	0.050 **	0.202 **	−0.199 **	0.174 **	0.152 **	0.082 **	0.218 **	0.153 **					
13	−0.067 **	−0.017	−0.095 **	−0.008	−0.134 **	0.171 **	−0.113 **	−0.083 **	−0.045 **	−0.125 **	−0.085 **	−0.273 **				
14	−0.057 **	−0.022 *	−0.052 **	−0.006	−0.080 **	0.101 **	−0.072 **	−0.052 **	−0.029 **	−0.065 **	−0.053 **	−0.176 **	0.134 **			
15	0.041 **	0.037 **	0.073 **	0.029 **	0.105 **	−0.086 **	0.077 **	0.095 **	0.053 **	0.111 **	0.082 **	0.192 **	−0.241 **	−0.170 **		
16	0.170 **	0.113 **	0.192 **	0.104 **	0.196 **	−0.080 **	0.142 **	0.094 **	0.105 **	0.238 **	0.147 **	0.234 **	−0.216 **	−0.166 **	−0.086 **	
17	−0.005	0.043 **	0.059 **	0.007	−0.027 **	0.168 **	−0.055 **	−0.010	0.065 **	0.024 **	0.025 **	−0.076 **	0.108 **	0.061 **	−0.050 **	0.027 **

^1^ Note: (1) Examine HW, (2) Supervise HW, (3) Finish HW together, (4) Assign HW, (5) Patiently, (6) Angry, (7) Read, (8) Discuss TV, (9) Set time, (10) Communicate, (11) Buy books, (12) Conflict with child, (13) Scold child, (14) Treat cool, (15) Talk with child, (16) Help analyzing, (17) Conflict with child. * *p* < *0*.05; ** *p* < *0*.01; Pearson correlations are reported for the four-point scale; Spearman correlations are reported for correlations between binary variables or between a binary variable and a four-point variable.

**Table 3 behavsci-12-00237-t003:** Latent class model comparisons.

Model	LL	AIC	BIC	Entropy	LMRA	BLRT	Average Latent Class Probabilities				
1-class	−206463	412985	413200	--	--	*p* < 0.0001	1.000				
2-class	−197010	394114	394463	0.759	*p* < 0.0001	*p* < 0.0001	0.925	0.932			
3-class	−194634	389399	389882	0.749	*p* < 0.0001	*p* < 0.0001	0.884	0.884	0.877		
4-class	−193541	387248	387865	0.760	*p* = 0.0048	*p* < 0.0001	0.854	0.838	0.867	0.874	
5-class	−192709	385619	386371	0.741	*p* = 0.0160	*p* < 0.0001	0.856	0.813	0.843	0.808	0.864

Note: LL = log-likelihood; BLRT = bootstrapped likelihood ratio test; LMRA = Lo–Mendell–Rubin adjusted test.

**Table 4 behavsci-12-00237-t004:** Summary of the discriminant analysis classification (four-class model).

	Predicted Group Membership			
Latent Class	Permissive	Supportive	Neglectful	Restrictive
Permissive	6717	5	2	77
Supportive	252	2306	0	18
Neglectful	240	0	1856	107
Restrictive	167	1	65	762
Classification accuracy	91.1%	99.7%	96.5%	79.0%

## Data Availability

The data presented in this study are available on request from the corresponding author.

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
