# Peer review of "Latent Class Analysis to Identify Parental Involvement Styles in Chinese Children’s Learning at Home"

_behavsci, 2022, doi:10.3390/bs12070237_

Round 1
Reviewer 1 Report
The authors offer an innovative and original reserach. Research questions should be included in the introduction section. A 'Conclusion' section should also be included as well after Discussion.
Please see attachement.

Author Response
Authors Response to Reviewers
Thank you for the opportunity to reply to the review comments.
We authors wish to re-emphasize the focus and purpose of our article:
Across the last several decades, past studies of the impact of parental involvement on student achievement showed mixed results, in part because of different definitions of parental involvement. This study showcased a methodology of latent class analysis (LCA), discriminant analysis, MANOVA, and correlation analysis, to determine and validate subpopulations of distinct types of parent involvement.
Our Focus: Clarify definition of parental involvement, as past studies with different definitions of parent involvement yielded mixed results.
Our Purpose: Provide a clear methodology, latent class analysis, used in other research areas to define sub-populations within a global population. We determined four distinct types of parent involvement by using this method.
Significance: Researchers using our four distinct types of parent involvement will obtain research findings suitable for comparisons.
Responses to Reviewer 1
Point 1. What about participants' consent to participate in the study? Did they sign any written consent form?
Author reply: The data used in the analysis were collected from parents responding to questions on a questionnaire. In the questionnaire instructions, we specified (a) potential benefits, discomfort or risks (b) the voluntary choice of respondent to take part in the survey, (c) that the information that they provide will be treated confidentially, and (d) that their responses will be reported anonymously. In our culture, survey participants typically do not sign consent forms, and our participants did not sign them. We have made a supplement on the manuscript:
“ The data used in the analysis were reported by parents. Stratified sampling yielded a representative sample of 12,575 7th and 8th grade students in a city in central China. This city's seventeen administrative areas cluster into four groups with similar gross domestic product (GDP), education spending, mean score on high-school entrance examinations, urban-rural distribution, resident population, and total population. From each of the four groups, we selected 53 junior high schools based on school type, school size, school educational quality, and the number of schools within that area. Within each of these schools, we sampled two or three 7th and 8th grade classes with diverse academic achievements (according to midterm test scores and class labels, such as low- and high-ability classes, and/or talented classes). All the students and their parents in the sampled classes participated in this project, resulting in a total sample of 12,575 students (53% boys and 47% girls; 52% 7th grade students and 48% 8th grade students) and parents. Sampled students’ parents responded to questions on a questionnaire voluntarily.”
Point 2: It seems that a conclusion section should be included after 'Discussion'. Or the author can reorganized 'Discussion' and name it Discussion and Conclusions' and extend it.
Author reply: We agree that the section, Discussion, should be re-titled, Discussion and Conclusion.We agree that the section, Discussion, should be re-titled, Discussion and Conclusion. The references in the introduction, discussion and conclusions were updated, the future study was extended.
We have added a few relevant articles in the last 1-5 years. We linked their findings to our Conclusions and Discussion section.
Chen, B.-B., Wiium, N., Dimitrova, R., Chen, N. (2019). The relationships between family, school and community support and boundaries and student engagement among Chinese adolescents. Current Psychology, 2019, 38, 705–714. https://doi.org/10.1007/s12144-017-9646-0
Xiong, Y., Qin, X., Wang, Q. et al. Parental Involvement in Adolescents’ Learning and Academic Achievement: Cross-lagged Effect and Mediation of Academic Engagement. Journal of Youth Adolescence, 2021, 50, 1811–1823. https://doi.org/10.1007/s10964-021-01460-w
Cosso, J., von Suchodoletz, A., Yoshikawa, H.. Effects of parental involvement programs on young children’s academic and social–emotional outcomes: A meta-analysis. Journal of Family Psychology, 2022, Advance online publication. http://dx.doi.org/10.1037/fam0000992
Ribeiro, L. M., Cunha, R. S., Silva, M. C. A. E., Carvalho, M., Vital, M. L.. Parental involvement during pandemic times: Challenges and opportunities. Education Sciences, 2021, 11(6), 302.
Huang, H., Hwang, G. J.. Facilitating Inpatients’ Family Members to Learn. Journal of Educational Technology & Society, 2019, 22(3), 74-87.
Wang, X., Xing, W.. Exploring the influence of parental involvement and socioeconomic status on teen digital citizenship: A path modeling approach. Journal of Educational Technology & Society, 2018, 21(1), 186-199.
Papadakis, S., Zaranis, N., Kalogiannakis, M.. Parental involvement and attitudes towards young Greek children’s mobile usage. International Journal of Child-Computer Interaction, 2019, 22, 100-144.
Future Research
“The current findings showed that parental involvement styles in children’s learning at home can be identified and categorized into four different latent class styles. These results suggest that only by identifying and modeling parental involvement styles at home, can researchers address research questions concerning parenting and students' academic achievement. Future studies could include questions related to parenting involvement with children that complete homework on electronic devices. The use of the four parenting styles would also permit statistical analyses with parent gender, subject area, and course type variables. Furthermore, this study showed how parent education and family income (components of family social economic status, SES) are related to parental involvement, so future studies can examine this relation and its possible causal mechanisms [6].
As this study indicates links between parental involvement and student academic achievement, future studies can discern its causal mechanisms and mediating variables, such as students' emotions or parental attachment [42]. Future research can explore how parent interactions in schools and with teachers and parental involvement styles with their children mutually influence each other [43]. Future studies can also examine these relations in other countries. Clarification of the definition of parenting involvement types delineated in this study facilitates future research on parenting involvement (e.g., comparisons across demographic attributes.)”
Reviewer 2 Report
This is an interesting paper in which authors provide an study related to the parental involvment in students bahvior and learning proccess.
Before considering the paper this reviewer suggest the following changes to be considered:
- First of all, first section related to introduction must be rewritten in depth considering for example if there is involved digital competence or technology such us augmented reality or other one.
- Moreover, the study is in general but, you compare the complete performance of the course? On a semester? IS there any difference by semester? Adn by gender? Or evenmore by subject? I mean the parents involvement is the same along the course and even in every subject?
- Related to the conclussions as there exist a lack of references in the introduction is complicated to consider it supported so I urge the authors to add more information by other authors.
- Furthermore, I urge the authors to add a section related to future work in which they consider the future using also technologies in the study.
- Finally, references are clearly too old, just only 1-2 references in the last five years, so please consider to indotruce the changes by considering new references as, for example, the following ones,as the journal recommedn that an important part of the references should be in the last 5 years:
- Wang, X., & Xing, W. (2018). Exploring the influence of parental involvement and socioeconomic status on teen digital citizenship: A path modeling approach. Journal of Educational Technology & Society, 21(1), 186-199.
- Ribeiro, L. M., Cunha, R. S., Silva, M. C. A. E., Carvalho, M., & Vital, M. L. (2021). Parental involvement during pandemic times: Challenges and opportunities. Education Sciences, 11(6), 302.
- Marín Suelves, D., Cuevas Monzonís, N., Gabarda Méndez, V. Digital competence for citizen: Analysis of trends in education (2021) RIED-Revista Iberoamericana de Educacion a Distancia, 24 (2), pp. 329-349.
- Huang, H., & Hwang, G. J. (2019). Facilitating Inpatients’ Family Members to Learn. Journal of Educational Technology & Society, 22(3), 74-87.
- Papadakis, S., Zaranis, N., & Kalogiannakis, M. (2019). Parental involvement and attitudes towards young Greek children’s mobile usage. International Journal of Child-Computer Interaction, 22, 100144.
- Vidal, I.M.G., López, B.C., Otero, L.C. New digital skills in students empowered with the use of Augmented Reality. Pilot Study (2021) RIED-Revista Iberoamericana de Educacion a Distancia, 24 (1), pp. 137-157.
- Anastasiou, S., & Papagianni, A. (2020). Parents’, teachers’ and principals’ views on parental involvement in secondary education schools in Greece. Education Sciences, 10(3), 69.
Author Response
Authors Response to Reviewer
Thank you for the opportunity to reply to the review comments.
We authors wish to re-emphasize the focus and purpose of our article:
Across the last several decades, past studies of the impact of parental involvement on student achievement showed mixed results, in part because of different definitions of parental involvement. This study showcased a methodology of latent class analysis (LCA), discriminant analysis, MANOVA, and correlation analysis, to determine and validate subpopulations of distinct types of parent involvement.
Our Focus: Clarify definition of parental involvement, as past studies with different definitions of parent involvement yielded mixed results.
Our Purpose: Provide a clear methodology, latent class analysis, used in other research areas to define sub-populations within a global population. We determined four distinct types of parent involvement by using this method.
Significance: Researchers using our four distinct types of parent involvement will obtain research findings suitable for comparisons.
Responses to Reviewers
Introduction Section
Point 1. First of all, first section related to introduction must be rewritten in depth considering for example if there is involved digital competence or technology such as augmented reality or other one.
Author reply: Our research did not involve the use of variables (questions) related to digital competence or technology. The questions were related to parenting style for the 17 indicator variables. Some questions did involve TV and games, but our main focus was on homework. A future study could include parenting involvement with children when using computers, tablets, phones, etc.
Point 2. Moreover, the study is in general but, you compare the complete performance of the course? On a semester? IS there any difference by semester? And by gender? Or even more by subject? I mean the parents involvement is the same along the course and even in every subject?
Author reply: These are important limitations of our study and suitable lines of future research. The focus of our study was to clarify and determine parenting involvement types. Future studies using these four distinct parenting involvement types can make comparisons by subject, gender, or course as you suggested.
The appropriate methodology for such a comparison is to (a) use one data set to create the classes, as we did in this study, and then (b) use another data set to make comparisons across classes, as future studies can do.
Point 3. Related to the conclusions as there exist a lack of references in the introduction is complicated to consider it supported so I urge the authors to add more information by other authors.
Finally, references are clearly too old, just only 1-2 references in the last five years, so please consider to introduce the changes by considering new references as, for example, the following ones, as the journal recommend that an important part of the references should be in the last 5 years:
Author reply: We agree that references in the introduction should be linked to our conclusions. We feel some of the older references are necessary to lay the foundation for our study. We have added a few relevant articles in the last 1-5 years. We linked their findings to our Conclusions and Discussion section.
Chen, B.-B., Wiium, N., Dimitrova, R., Chen, N. (2019). The relationships between family, school and community support and boundaries and student engagement among Chinese adolescents. Current Psychology, 2019, 38, 705–714. https://doi.org/10.1007/s12144-017-9646-0
Xiong, Y., Qin, X., Wang, Q. et al. Parental Involvement in Adolescents’ Learning and Academic Achievement: Cross-lagged Effect and Mediation of Academic Engagement. Journal of Youth Adolescence, 2021, 50, 1811–1823. https://doi.org/10.1007/s10964-021-01460-w
Cosso, J., von Suchodoletz, A., Yoshikawa, H.. Effects of parental involvement programs on young children’s academic and social–emotional outcomes: A meta-analysis. Journal of Family Psychology, 2022, Advance online publication. http://dx.doi.org/10.1037/fam0000992
Ribeiro, L. M., Cunha, R. S., Silva, M. C. A. E., Carvalho, M., Vital, M. L.. Parental involvement during pandemic times: Challenges and opportunities. Education Sciences, 2021, 11(6), 302.
Huang, H., Hwang, G. J.. Facilitating Inpatients’ Family Members to Learn. Journal of Educational Technology & Society, 2019, 22(3), 74-87.
Wang, X., Xing, W.. Exploring the influence of parental involvement and socioeconomic status on teen digital citizenship: A path modeling approach. Journal of Educational Technology & Society, 2018, 21(1), 186-199.
Papadakis, S., Zaranis, N., Kalogiannakis, M.. Parental involvement and attitudes towards young Greek children’s mobile usage. International Journal of Child-Computer Interaction, 2019, 22, 100-144.
Point 4 Furthermore, I urge the authors to add a section related to future work in which they consider the future using also technologies in the study.
Author reply: We agree that this last paragraph be given the title, Future Research, and include comments about expanding questions to include technology use by children and making comparisons between parent gender, course or subject levels:
Future Research
“The current findings showed that parental involvement styles in children’s learning at home can be identified and categorized into four different latent class styles. These results suggest that only by identifying and modeling parental involvement styles at home, can researchers address research questions concerning parenting and students' academic achievement. Future studies could include questions related to parenting involvement with children that complete homework on electronic devices. The use of the four parenting styles would also permit statistical analyses with parent gender, subject area, and course type variables. Furthermore, this study showed how parent education and family income (components of family social economic status, SES) are related to parental involvement, so future studies can examine this relation and its possible causal mechanisms [6].
As this study indicates links between parental involvement and student academic achievement, future studies can discern its causal mechanisms and mediating variables, such as students' emotions or parental attachment [42]. Future research can explore how parent interactions in schools and with teachers and parental involvement styles with their children mutually influence each other [43]. Future studies can also examine these relations in other countries. Clarification of the definition of parenting involvement types delineated in this study facilitates future research on parenting involvement (e.g., comparisons across demographic attributes.)”
Round 2
Reviewer 2 Report
Paper now is ready for publication.